

# Modulation of stress response and productive performance of *Litopenaeus vannamei* through diet

Eliza M. Martínez-Antonio, Ilie S. Racotta, Juan C. Ruvalcaba-Márquez and Francisco Magallón-Barajas

Programa de Acuicultura, Centro de Investigaciones Biológicas del Noroeste S. C., La Paz, B.C.S., Mexico

Corresponding author
Ilie S. Racotta, iracotta@cibnor.mx

## ABSTRACT

The high tolerance of *Litopenaeus vannamei* to a wide range of salinity (1–50 psu) makes this species an excellent candidate for culture under low salinity, decreasing shrimp epidemics and water pollution in some coastal areas. However, salinity levels outside the optimal range could impose several physiological constraints that would in turn affect growth and survival, particularly in the presence of additional stressors (e.g. high densities, handling practices, and hypoxia). Despite shrimp susceptibility to individual stressors has been widely addressed, information regarding response to chronic and acute stressors combined and its relation to diet is scarce. Thus, the aim of our study was to determine the effect of diet on the susceptibility to chronic (low salinity) and acute (hypoxia and escape response) stressors in terms of culture performance and physiological indicators. We evaluated overall performance during culture of *L. vannamei* at low salinity (6 psu), fed with an experimental diet with low protein and high carbohydrate content (26% protein and 6% fish meal plus probiotic mixture) and compared to a commercial formula with high protein and low carbohydrate content (40% crude protein and 20% fish meal without probiotic mixture). At the end of the rearing experiment, shrimp were exposed to two types of acute stress, hypoxia and escape. Biochemical (hemocyanin, total proteins, glucose, and lactate) and bioenergetic (adenylic energy charge and arginine phosphate levels) variables were measured to assess chronic stress response (salinity) and acute stress response (hypoxia or escape). The experimental diet resulted in higher muscle energy status that was not affected by low salinity, although lipid levels were lower under this condition. This diet partially counteracted the low performance at low salinity and promoted greater protein efficiency. Hypoxia induced strong hyperglycemic and lactate increase as response, whereas escape response was characterized by a depletion of arginine phosphate levels, with a stronger decrease in shrimp fed experimental diet, due to the high initial level of this reserve. Some data (glucose levels in hemolymph and lipids in hepatopancreas) suggest that shrimp under chronic stress conditions (low salinity and high densities) present a low ability to respond to subsequent acute stressors such as hypoxia or escape. This work indicates that diet can increase the energy status of shrimp, enabling them to overcome potential multifactorial stressors, which are common in farming systems.

## INTRODUCTION

Shrimp culture has been one of the major growth areas in worldwide aquaculture during the last 55 years, a production of 4.156 million tons was reached in 2016 (*FAO, 2018*). The extent of this industry is mainly on the basis of formulated feeds, which are manufactured with high contents of protein and fish meal to promote rapid shrimp growth (*Shiau, 1998*; *Cuzon et al., 2004*) with a poor consideration of environmental impact, production costs (increasing prices of fish meal and oil), and shrimp responses to several stressors inherent to the current culture systems (*Tacon & Metian, 2008*). Thus, there is a great need to formulate functional feeds that, in addition to providing good culture yield, enhance the capacity of shrimp to minimize stress.

The high tolerance of *Litopenaeus vannamei* to a wide range of salinity (1–50 psu), (*Pante, 1990*) makes this species an excellent candidate for culture under low salinity in inland farming, developed with the aim of decreasing shrimp epidemics and water pollution in some coastal areas (*Li et al., 2017*). However, salinity levels outside the optimal range and prolonged exposure could impose several physiological constraints that would in turn affect growth and survival. This is particularly the case for culture under low salinities, where hyper-osmoregulation implies high energy demand, and therefore nutritional requirements of shrimp grown at low salinities has been extensively studied (for reviews see *Romano & Zeng, 2012*; *Li et al., 2017*). Although proteins plays an important role as energy source for osmoregulation, the benefits as enhancers of growth and survival at low salinity are controversial (*Li et al., 2017*). In contrast, a sparing effect of protein could be suggested when intermediate levels of carbohydrates are included in shrimp diet at low salinity (*Wang et al., 2014*; *Wang et al., 2015*). Finally, essential lipids such as phospholipids and cholesterol, as well as supplementation of minerals resulted in better performance of shrimp at low salinity (*Gong et al., 2004*). In addition, supplementation of commercial probiotics increased survival but not growth in shrimp cultured at 2 psu salinity (*Li et al., 2009*). Supplementation of *Lactobacillus plantarum* improved resistance to low salinity stress test (*Zheng et al., 2017*).

In intensified systems, hypoxia and shrimp handling increase given the high densities. These increased factors are considered stressors since these affect shrimp physiology, causing a reduction of culture yield in terms of growth and survival. The response to stress consists in the mobilization of energy substrates (amino acids, glucose, triglycerides, among others) to produce enough energy to meet such factors (*Lucas, 1996*).

There are a number of researches regarding shrimp response to prolonged exposure (chronic stress) to low salinity (*Rosas et al., 2001a*; *Li et al., 2017*) and/or temporary events (acute stress) such as hypoxia and handling (*Racotta & Palacios, 1998*; *Racotta, Palacios & Méndez, 2002*; *Aparicio-Simón et al., 2010*). However, there are no studies addressing the combined effect of chronic and acute stress on shrimp capacity to overcome such stressors in terms of energy regulation through diet. Thus, the aim of our study was to determine the effect of diet on the susceptibility to chronic salinity followed by acute exposure to hypoxia and escape, in terms of culture performance and energy status.

## MATERIALS & METHODS

### Salinity reduction and handling of organisms

*Litopenaeus vannamei* postlarvae were obtained from the private company Acuacultura Ma hr S.A. de C.V. (La Paz, Mexico). Postlarvae were fed with commercial feed (40% protein level) and acclimated to experimental conditions in the experimental nutrition laboratory of the Mexican research center (Centro de Investigaciones Biologicas del Noroeste (CIBNOR)) for approximately 6 weeks.

Prior to the transfer of juveniles ($2.08 \pm 0.35$ g) to the experimental units, half of the stock was acclimated to well water at low salinity (6 psu). For this purpose, 400 shrimp were stocked in 500-L polyethylene tanks at low depth (0.3 m) and bottom area of 3.1 m$^2$. During seven days, salinity was reduced (5 psu/day) until the desired salinity was reached (*Ponce-palafox, Martinez-palacios & Ross, 1997*). Shrimp ($2.97 + 0.55$ g) were randomly distributed in 60 L aquariums at a density of 67 organisms/m$^2$.

During 42 days, shrimp were cultured at low ($6.2 \pm 0.03$ psu, well water) and high ($37.3 \pm 0.05$ psu, sea water) salinities under laboratory conditions ($25.9 \pm 0.1$ °C, dissolved oxygen $= 6.3 \pm 0.1$ mgO$_2$.L$^{-1}$, and photoperiod $= 12$h: 12 h light: dark). Water was exchanged (80–90%) twice a week. Two diets were tested: control and experimental, as specified hereafter, using four replicates for each diet-salinity combination and with 15 shrimp for each replicate. The feeding rate was 5% of the total biomass in four daily rations for each aquarium (9:00 h, 12:00 h, 15:00 h and 17:00 h).

The following zootechnical parameters were obtained:

Weight gain (%) (WG) = ((final weight (g) –initial weight (g))*100)/ initial weight (g); Specific growth rate (%) (SGR) = 100 * (ln final weight (g) –ln initial weight (g))/days of experiment; Survival (%) = (shrimp initial number –dead shrimp number)/shrimp initial number ×100; Feed conversion ratio (FCR) = feed intake (dry matter) (g)/weight gain (g); Protein efficiency ratio (PER) = wet weight gain (g) / dry protein intake (g); and

Productive performance = (Survival * SGR)/FCR.

### Feed and proximal composition

The control diet was prepared in accordance with a commercial formula that consisted in high inclusion of fish meal (20%). The experimental diet was formulated with low inclusion of fish meal (6%) and high phosphorus (1.5%), a mixture of free amino acids, vitamins, minerals, cholesterol and Butylated hydroxytoluene (BHT). This diet was adjusted to shrimp nutritional requirements, as reported in scientific research until 2015 (Table 1). In addition, a selection of yeasts and lactobacilli in liquid medium was included in the experimental diet. Once extruded and dried, the experimental diet was subjected to baths of organic acids and oil with astaxanthin, respectively. Both diets, experimental and control, were prepared with the same sources of protein, carbohydrates, and lipids. The proximate composition of control and experimental diets is shown in Table 2.

Fatty acids were analyzed for both diets according to *Palacios et al. (2005)*. Lipids were extracted according to *Folch, Lees & Sloane-Stanley (1957)*, boron trifluoride-methanol (BF3–14% methanol, 3-3021; Sigma, St. Louis, MO, USA) was used for hydrolysis

**Table 1   Composition of control and experimental diets.**

| Ingredient | Control | Experimental |
|---|---|---|
| Fish meal | 20 | 6 |
| Soybean meal | 37.2 | 24.1 |
| Wheat meal | 35 | 44.5 |
| Fish oil | 4 | 2.6 |
| Soy lecithin | 3 | 5 |
| Alginate acid | 0.75 | 2 |
| Vitamin Mix | 0.09[a] | 0.27[c] |
| Mineral Mix | 0.05[a] | 8.5[d] |
| Yeast[b] | 0 | 4 |
| Probiotic Mix[e] | 0 | 1.2 |
| Organic acids Mix[f] | 0 | 1 |
| Fish oil + Astaxanthin | 0 | 0.5 |
| Free amino acids mix[g] | 0 | 0.4 |

**Notes.**
Data expressed in dry weight pecentage of diets
[a] Commercial mix from PIASA, SA de CV.
[b] Commercial yeast (*Saccharomyces cerevisiae*).
[c] Laboratory vitamins mix: Choline, Butylated hydroxytoluene (BHT), cholesterol, Vitamin A, C, B1, B2, B3, B5, B6, B7, B8, B9, B12, B20, D, E, and K.
[d] Laboratory minerals mix: $NaH_2PO_4$, $CaH_2PO_4$, $KH_2PO_4$, $MgSO_4H_2O$, $ZnSO_4 \cdot 7H_2O$, $ZnSO_4 \cdot H_2O$, $MnSO_4 \cdot 4H_2O$, $FeSO_4$, $CuSO_4 \cdot 7H_2O$, $Na_2SeO_3$, KI, $Na_2MoO_4$.
[e] Laboratory strain mix of *Wickerhamomyces anomalus*, *Pichia kudriavzevii*, *Lactobacillus plantarum* and *Bacillus subtilis*.
[f] Organic acids Mix: propionic, butyric acetic and nicotinic acids.
[g] Laboratory free amino acids: arginine, methionine, lysine, tryptophan, and threonine.

**Table 2   Proximate analysis of control and experimental diets.** Data expressed as mean ± standard error Values with different letters in the same row present significant differences ($P < 0.05$).

| | Control | Experimental |
|---|---|---|
| Crude protein (%) | $40.4 \pm 0.2^a$ | $26.1 \pm 0.1^b$ |
| Lipid (%) | $8.1 \pm 0.1^a$ | $9.0 \pm 0.05^b$ |
| Nitrogen-free extract (%) | $43.3^a$ | $54.1^b$ |
| Water content (%) | $6.4 \pm 0.1^a$ | $4.7 \pm 0.1^b$ |
| Nitrogen (%) | $6.5 \pm 0.02^a$ | $4.2 \pm 0.02^b$ |
| Crude fiber (%) | $1.0 \pm 0.1^a$ | $0.6 \pm 0.1^b$ |
| Ashes (%) | $7.2 \pm 0.04^a$ | $10.2 \pm 0.02^b$ |
| Gross energy (cal/g) | $4809.6 \pm 3.1^a$ | $4391.4 \pm 1.4^b$ |

and esterification of fatty acids. The resulting methyl esters were separated by gas chromatography (G890N; Agilent Technologies, Santa Clara, CA, USA) with a DB-23 silica column (30m × 0.25mmID × 0.25mm) film thickness), helium as carrier gas, a temperature ramp of 110–220 °C, and a flame ionization detector. An internal standard (23:0) was used to identify fatty acids in terms of concentration of each fatty acid corrected. Fatty acid composition of both diets is presented in Table 3.
**Table 3  Fatty acids of control and experimental diets.**

| Fatty acid (%) | Control | Experimental |
|---|---|---|
| 16:0 | 15.54 ± 0.08[a] | 14.40 ± 0.09[b] |
| 18:0 | 4.20 ± 0.03[a] | 3.92 ± 0.02[b] |
| 16:1n-7 | 3.96 ± 0.01[a] | 3.51 ± 0.02[b] |
| 18:1n-9 | 13.44 ± 0.04[a] | 14.64 ± 0.02[b] |
| 18:1n-7 | 2.57 ± 0.02[a] | 2.23 ± 0.01[b] |
| 20:1n-9 | 1.64 ± 0.01[a] | 1.53 ± 0.003[b] |
| 22:1n-11 | 2.30 ± 0.04[a] | 2.17 ± 0.03[a] |
| 18:2n-6 | 27.69 ± 0.03[a] | 31.44 ± 0.14[b] |
| 18:3n-3 | 3.84 ± 0.02[a] | 4.32 ± 0.01[b] |
| 18:4n-3 | 1.09 ± 0.01[a] | 1.00 ± 0.01[b] |
| 20:4n-6 | 0.43 ± 0.001[a] | 0.36 ± 0.01[b] |
| 20:5n-3 (EPA) | 6.81 ± 0.06[a] | 6.15 ± 0.06[b] |
| 22:5n-6 | 1.28 ± 0.01[a] | 1.16 ± 0.02[b] |
| 22:6n-3 (DHA) | 8.27 ± 0.10[a] | 7.33 ± 0.18[b] |
| Σ SAT | 23.96 ± 0.11[a] | 21.74 ± 0.13[b] |
| Σ MUFA | 25.86 ± 0.12[a] | 25.75 ± 0.01[a] |
| Σ PUFA | 50.19 ± 0.23[a] | 52.50 ± 0.13[b] |
| Σ HUFA | 18.32 ± 0.18[a] | 16.41 ± 0.28[b] |
| Fatty acids total (mg/g) | 8.13 ± 0.31[a] | 8.75 ± 0.38[a] |

**Notes.**

Data expressed as means ± standard error.

Σ SAT, sum of saturated fatty acids; Σ MUFA, sum of monounsaturated fatty acids; Σ PUFA, sum of polyunsaturated fatty acids; Σ HUFA, sum of highly unsaturated fatty acids; $\Sigma\, n-6$, sum of $n-6$; $\Sigma\, n-3$, sum of $n-3$.

Values with different letters in the same row present significant differences ($P < 0.05$).

## Stress tests and biochemical analysis

Stress tests were divided in three groups: hypoxia, escape response, and control. Ten shrimp were considered for each group and salinity-diet combination. The shrimp were fasted for 15 h prior to the stress test. For hypoxia, dissolved oxygen was decreased by nitrogen bubbling to $1.0 \pm 0.5$ mg L$^{-1}$ for 30 min. The escape response (tail-flipping) was induced by prodding shrimp until exhaustion (around 30 s), which is characterized by a prolonged unwillingness to respond by tail-flipping (*Robles-Romo, Zenteno-Savín & Racotta, 2016*). Shrimp of the control (baseline) group were undisturbed and maintained at normoxia ($5.6 \pm 0.4$ mg L$^{-1}$). Immediately (less than one min) after the end of the application of the stressor applied, 100–200 µl of hemolymph were collected in each shrimp from the ventral sinus at the base of the first abdominal segment, this procedure was performed using a cooled-anticoagulant solution formulated with 5% sodium oxalate in isotonic saline (*Mendoza, 1992*). Samples were immediately frozen in liquid nitrogen and stored at 75 °C for further analyses. The same sampling procedure was applied for control shrimp to obtain baseline values in the absence of stress.

Hemolymph was centrifuged at 1,350 g for 10 min at 4 °C, plasma was collected for quantification of hemocyanin, total proteins, glucose, and lactate. The hepatopancreas and

muscle (first abdominal segment) were dissected, lyophilized, grinded, re-hydrated and homogenized for quantification of levels of total proteins, total lipids, and triglycerides.

Commercial kits were employed to determine lactate (PAP, Randox, U. K.), glucose (GOD-PAP; Boehringer Mannheim GmbH, Mannheim, Germany), and triglycerides (GPO-PAP, Randox), methods were adapted to microplates (*Palacios et al., 1999*).

Plasma was 1:100 with saline isotonic solution (450 mM NaCl and 10 mMKCl). Total proteins were determined according to *Bradford (1976)* using a commercial reagent concentrate (500-0006; Bio-Rad) and bovine serum albumin (A-3912; Sigma) as standard. Total proteins were read on a microplate reader at 595 nm (Multiscan GO; Thermo Fisher Scientific, Waltham, MA, USA). In hepatopancreas and muscle homogenates, total protein was determined after digestion with NaOH (0.1N) at dilution of 1:20.

Hemocyanin was measured directly from 10-μl plasma diluted 1:20 with saline isotonic solution. Absorbance was read at 335 nm and concentrations were calculated using an extinction coefficient ($E^{1\%}$) of 2.83 for shrimp hemocyanin (*Hagerman, 1986*). Total lipids were determined by the sulphophosphovanillin method (*Barnes & Blackstock, 1973*).

Abdominal muscle was dissected and homogenized under cryogenic conditions. Extraction and analysis of adenylic nucleotides and arginine phosphate were performed according to *Robles-Romo, Zenteno-Savín & Racotta (2016)*. Extraction consisted on homogenization in 10% trichloroacetic acid and neutralization by means of a mixture of trioctylamine and dichloromethane (1:5 v/v). Nucleotides were separated by ion pairing reverse phase high-performance liquid chromatography (HPLC) (model 1100; Agilent Technologies, Santa Clara, CA, USA) with an octadecylsilane C18 column (Hyper Clone 150 mm length, 4.6 mm internal diameter, 3 μm particle size diameter; Phenomenex, Torrance, CA, USA) and a security guard cartridge C18 (40 mm length, 3.0 mm internal diameter, Phenomenex). Conditions of this procedure were the following: a flow rate of 0.8 mL min$^{-1}$ using a mobile phase of 0.15 M $NaH_2PO_4$ buffer, 3 mM of tetrabutylammonium as the ion-pairing agent, and 8% methanol adjusted to pH 6.0 with 5 N NaOH. The adenylic energy charge (AEC) was calculated in accordance with *Atkinson (1968)*: AEC= ATP+1/2ADP/ATP+ADP+AMP.

## Statistical analysis

Homoscedasticity of variances and normality of data were verified. A Two-factor analysis of variance (ANOVA) (2x2) was employed to determine the effect of salinity (6 and 37 psu) and diet (control and experimental) over the zootechnical parameters in terms of weight gain, SGR, PER, FCR, survival, and productive performance. A trifactorial ANOVA ($2 \times 2 \times 3$) was performed to determine the influence of salinity (6 and 37 psu), diet (control and experimental) and stress factor (hypoxia, escape response, and baseline) over metabolic variables in hemolymph, muscle, and hepatopancreas. Only when a significant triple interaction was observed were individual means for each salinity-diet or salinity-diet-stress combination compared (Tukey's HSD test), and differences are indicated in the tables. Otherwise, global means within each factor or two-factor combination are mentioned and compared in the text, together with the corresponding significant main effect of this factor or the two-factor interaction. The software used was STATISTICA

**Table 4 Biological performance (mean ± standard error) of juveniles of *L. vannamei* reared at 37 and 6 psu, fed with control and experimental diets for 6 weeks.**

| Salinity | 37 psu | | 6 psu | | ANOVA | | |
|---|---|---|---|---|---|---|---|
| Diet | Control | Experimental | Control | Experimental | Salinity | Diet | SxD |
| WG (%) | 304 ± 8$^a$ | 246 ± 6$^b$ | 253 ± 7$^b$ | 274 ± 5$^b$ | NS | **<0.05** | <0.01 |
| SGR (%/day) | 2.5 ± 0.1$^a$ | 2.2 ± 0.05$^b$ | 2.2 ± 0.1$^b$ | 2.4 ± 0.04$^{ab}$ | NS | NS | <0.01 |
| FCR | 1.9 ± 0.1$^b$ | 2.1 ± 0.1$^{ab}$ | 2.5 ± 0.2$^a$ | 2.1 ± 0.1$^{ab}$ | NS | NS | <0.05 |
| Survival (%) | 91.8 ± 5.8$^{ab}$ | 93.7 ± 3.9$^{ab}$ | 72.2 ± 6.5$^b$ | 98.1 ± 1.9$^a$ | NS | <0.05 | <0.05 |
| PER | 1.4 ± 0.1$^b$ | 1.9 ± 0.04$^a$ | 1.1 ± 0.01$^c$ | 1.9 ± 0.01$^a$ | NS | **<0.01** | <0.05 |
| Productive performance | 1.23 ± 0.15$^a$ | 1.0 ± 0.10$^{ab}$ | 0.7 ± 0.12$^b$ | 1.1 ± 0.05$^a$ | NS | NS | <0.01 |

**Notes.**

Results of two-way ANOVA are indicated in last columns (NS= not significant).

Following Tukey's HSD test, values with different letters in the same row present significant differences ($P < 0.05$).

WG, weight gain; SGR, specific growth rate; FCR, feed conversion rate; PER, protein efficiency ratio.

(version 10.0). Differences were considered significant al $P < 0.05$. Means with different letters are statistically different.

## RESULTS

### Productive performance

Shrimp growth in terms of WG ($F_{1,12} = 35.8$, $P = 0.00006$) and SGR ($F_{1,12} = 15.1$, $P = 0.0022$), was significantly affected by the interaction between salinity and diet, shrimp fed control diet and reared at 6 psu presented significantly lower values when compared to those reared at 37 psu and fed the same diet. Similarly, the lowest PER ($F_{1,12} = 6.4$, $P = 0.026$) and highest FCR ($F_{1,12} = 6.2$, $P = 0.029$) were detected in shrimp fed control diet and reared at 6 psu, therefore, these presented the lowest productive performance when compared with the rest of the treatments (Table 4).

In addition, as shown in Table 4, survival was higher in shrimp fed experimental diet than those fed control diet (main effect of diet $F_{1,12} = 8.0$, $P = 0.015$). However, such an effect is due to the significantly lower survival of shrimp reared at 6 psu and fed control diet ($72.2 ± 6.5\%$) (interaction, $F_{1,12} = 5.9$, $P = 0.031$).

### Biochemical responses

Shrimp grown at low salinity (6 psu) presented significantly lower levels glucose ($16.9 ± 1.4$ mg.dL$^{-1}$) when compared to shrimp at high (37 psu) salinity ($25.7 ± 2.5$ mg.dL$^{-1}$, main effect $F_{1,97} = 21.7$, $P = 0.00001$). Hemocyanin was also affected by salinity (main effect $F_{1,99} = 5.3$, $P = 0.023$); however, such effect was dependent on diet (interaction $F_{2,99} = 5.0$, $P = 0.0084$): differences between both salinities were observed only for the experimental diet (6 psu: $51.5 ± 3.8$ mg.mL$^{-1}$ and 37 psu: $69.1 ± 4.1$ mg.mL$^{-1}$, $P < 0.01$). Diet as single factor did not affect any variable in hemolymph, although several interactions were observed between diet and salinity or stress, as described furtherly.

Stress affected significantly all of the variables in hemolymph, both as single factor or combined with salinity and diet (Table 5). Particular effects of hypoxia and escape response were observed for the different variables analyzed. Protein levels were significantly lower after escape response ($84.1 ± 4.3$ mg.mL$^{-1}$) when compared to baseline and hypoxia
Table 5 Biological performance (mean ± standard error) of juveniles of *L. vannamei* reared at 37 and 6 psu, fed with control and experimental diets for 6 weeks.

| | 37 psu | | 6 psu | | S | D | st | SxD | Sxst | Dxst | Trt |
|---|---|---|---|---|---|---|---|---|---|---|---|
| | Control | Exp | Control | Exp | | | | | | | |
| Total Proteins (mg mL$^{-1}$) | | | | | NS | NS | ** | NS | NS | NS | NS |
| Baseline | 97.2 ± 11.6 | 127.1 ± 13.4 | 94.4 ± 19.6 | 105.1 ± 11.3 | | | | | | | |
| Hypoxia | 110.9 ± 15.2 | 118.2 ± 14.4 | 106.6 ± 12.1 | 121.8 ± 15.6 | | | | | | | |
| Escape | 78.1 ± 4.7 | 95.6 ± 11.5 | 99.2 ± 5.4 | 62.3 ± 4.6 | | | | | | | |
| Hemocyanin (mg mL$^{-1}$) | | | | | * | NS | NS | ** | * | ** | ** |
| Baseline | 52.8 ± 4.5[abc] | 73.7 ± 6.4[a] | 42.3 ± .5.7[bc] | 50.0 ± 2.7[abc] | | | | | | | |
| Hypoxia | 50.1 ± 3.1[abc] | 63.7 ± 6.3[ab] | 56.6 ± 6.1[abc] | 68.5 ± 7.5[ab] | | | | | | | |
| Escape | 56.2 ± 3.8[abc] | 70.2 ± 8.7[a] | 70.1 ± 3.1[a] | 36.0 ± 2.6[c] | | | | | | | |
| Glucose (mg dL$^{-1}$) | | | | | ** | NS | ** | NS | ** | * | NS |
| Baseline | 12.7 ± 1.3 | 18.8 ± 3.0 | 10.8 ± 2.6 | 14.5 ± 3.0 | | | | | | | |
| Hypoxia | 36.1 ± 3.0 | 53.5 ± 8.7 | 24.5 ± 4.4 | 28.6 ± 1.8 | | | | | | | |
| Escape | 18.6 ± 2.9 | 16.7 ± 2.3 | 15.5 ± 8.7 | 7.2 ± 0.7 | | | | | | | |
| Lactate (mg dL$^{-1}$) | | | | | NS | NS | ** | NS | NS | NS | NS |
| Baseline | 9.0 ± 0.6 | 6.6 ± 1.1 | 8.1 ± 0.4 | 8.4 ± 0.9 | | | | | | | |
| Hypoxia | 58.1 ± 11.3 | 49.6 ± 6.5 | 52.2 ± 7.4 | 68.4 ± 5.9 | | | | | | | |
| Escape | 21.3 ± 1.6 | 19.4 ± 2.1 | 19.5 ± 1.6 | 13.5 ± 1.8 | | | | | | | |

**Notes.**

Results of two-way ANOVA are indicated in last columns (NS, not significant).

Following Tukey's HSD test, values with different letters in the same row present significant differences ($P < 0.05$).

WG, weight gain; SGR, specific growth rate; FCR, feed conversion rate; PER, protein efficiency ratio.

($114.9 \pm 7.1$ and $106.4 \pm 6.9$ mg.mL$^{-1}$, respectively, stress main effect $F_{2,97} = 6.9$, $P = 0.0016$). Hemocyanin was not affected by stress as single factor, although all interactions were significant (Salinity $\times$ stress, $F_{2,99} = 4.3$, $P = 0.016$, Diet x stress, $F_{1,99} = 6.1$, $P = 0.0031$ and Salinity x Diet x stress, $F_{2,99} = 5.0$, $P = 0.0084$). For example, hemocyanin levels increased after escape response in shrimp grown at 6 psu and fed control diet, whereas a decrease (although not significant) was observed in those fed experimental diet at the same salinity, and no effect was observed at 37 psu for both diets (Table 5). Shrimp exposed to hypoxia presented significantly higher levels of glucose in hemolymph ($35.8 \pm 3.1$ mg.mL$^{-1}$) when compared to undisturbed or induced to escape shrimp ($14.3 \pm 1.3$ and $14.2 \pm 1.1$ mg dL$^{-1}$, respectively, stress main effect $F_{2,97} = 51.2$, $P < 0.00001$). Moreover, as denoted by the significant interaction, the influence of hypoxia was more pronounced at 37 psu than at 6 psu ($44.8 \pm 4.9$ vs. $26.7 \pm 2.2$ mg dL$^{-1}$, respectively, Salinity $\times$ stress $F_{2,97} = 5.4$, $P = 0.0061$) and in shrimp fed experimental diet with regard to those fed control diet ($40.4 \pm 2.4$ vs. $30.6 \pm 2.9$ mg dl$^{-1}$, respectively, Diet $\times$ stress $F_{2,97} = 5.5$, $P = 0.0056$). Lactate significantly increased in response to both stress conditions, although it was more pronounced at hypoxia ($57.3 \pm 4.0$ mg dL$^{-1}$) than at escape response ($18.3 \pm 1.0$ mg dL$^{-1}$), when compared to baseline values ($8.0 \pm 0.4$ mg dL$^{-1}$, stress main effect $F_{2,99} = 116.9$, $P < 0.00001$).

Lipid and triglyceride (TG) levels in hepatopancreas were significantly lower at 6 psu ($164.5 \pm 12.3$ and $43.9 \pm 3.4$ mg.g$^{-1}$, respectively) than at 37 psu ($204.5 \pm 7.8$

and $57.9 \pm 1.8$ mg.g$^{-1}$ salinity main effect $F_{1,103} = 7.9$, $P = 0.0059$ and $F_{1,103} = 18.4$, $P = 0.00004$, respectively). However, as reflected in the significant interaction, such effect was strongly dependent on diet, since it was detected for the control but not for the experimental diet (interaction $F_{1,103} = 9.0$, $P = 0.0034$ and $F_{1,103} = 11.3$, $P = 0.0011$, respectively). In addition, for lipids, the significantly lower values at low salinity with regard to high salinity were observed in shrimp submitted to hypoxia and escape stresses, as detected in the significant triple interaction (Table 6, $F_{2,103} = 3.1$, $P = 0.049$). For triglycerides, the influence of diet as single factor was significant, the highest levels were found in shrimp fed experimental diet ($59.0 \pm 2.3$ vs. $42.8 \pm 3.0$ mg.g$^{-1}$, $F_{1,103} = 24.4$, $P = 0.000003$). Protein levels in hepatopancreas were affected by salinity, however, this effect depended on diet and stress condition, as indicated by the significant interactions ($F_{1,103} = 6.6$, $P = 0.011$ and $F_{2,103} = 10.1$, $P = 0.0001$, respectively). Protein levels were higher in shrimp fed control diet at 6 psu ($329.9 \pm 20.8$ mg.g$^{-1}$) than those fed experimental diet at the same salinity ($266.2 \pm 15.9$ mg.g$^{-1}$), whereas similar levels for both diets were observed at 37 psu. A significant interaction was detected between salinity and stress given that protein levels increased with hypoxia in shrimp grown at 37 psu ($344$ vs. $\pm 24.9$ mg g$^{-1}$ vs. baseline values of $276.5 \pm 13.9$ mg. g$^{-1}$), whereas a reverse trend (although not significant) was observed at 6 psu (baseline: $304 \pm 28.4$ mg g$^{-1}$; hypoxia $251.6 \pm 15.8$ mg g$^{-1}$).

Lactate levels in muscle were significantly lower at 6 psu ($5.5 \pm 0.3$ mg.g$^{-1}$) than at 37 psu ($7.2 \pm 0.2$ mg.g$^{-1}$, $F_{1,102} = 30.3$, $P < 0.00001$), and affected by stress with a significant increase in the escape response ($7.6 \pm 0.3$ mg.g$^{-1}$ vs. baseline values $5.5 \pm 0.3$ mg.g$^{-1}$ and $6.0 \pm 0.3$ mg.g$^{-1}$ in shrimp exposed to hypoxia, $F_{2,102} = 16.7$, $P < 0.00001$). As shown by significant interaction salinity $\times$ diet, ($F_{1,102} = 4.2$, $P = 0.043$), shrimp fed experimental diet presented significantly higher levels of lactate in the muscle ($6.4 \pm 0.4$ mg.g$^{-1}$) with regard to those fed control diet ($4.5 \pm 0.4$ mg.g$^{-1}$), only at 6 psu. Protein levels in muscle significantly increased with both stressors (hypoxia $= 625.5 \pm 11.5$ mg.g$^{-1}$ and escape response $= 705.0 \pm 20.4$ mg.g$^{-1}$) when compared to baseline levels ($550.3 \pm 19.3$ mg.g$^{-1}$, stress main effect $F_{2,102} = 18.9$, $P < 0.00001$). In contrast, lipid levels in muscle significantly increased under escape stress ($49.0 \pm 2.5$ mg.g$^{-1}$) when compared to hypoxia and baseline values ($36.8 \pm 2.1$ and $33.1 \pm 2.4$ mg.g$^{-1}$, $F_{2,102} = 14.3$, $P < 0.00001$). Experimental diet significantly increased the total lipids in muscle ($42.9 \pm 2.2$ mg.g$^{-1}$), in comparison with control diet ($36.3 \pm 1.9$ mg.g$^{-1}$, diet effect $F_{1,102} = 7.3$, $P = 0.008041$).

On the other hand, Arg-P was significantly higher in shrimp fed experimental diet ($4.6 \pm 0.5$ $\mu$moles.g$^{-1}$) than in those fed control diet ($1.6 \pm 0.2$ $\mu$moles.g$^{-1}$, diet effect $F_{1,100} = 35.6$, $P < 0.00001$). Conversely, a decrease of Arg-P was observed following the escape response ($1.5 \pm 0.2$ $\mu$moles.g$^{-1}$) when compared to baseline values ($4.7 \pm 0.6$ $\mu$moles.g$^{-1}$), whereas a slight and non-significant decline was observed in shrimp submitted to hypoxia ($3.6 \pm 0.7$ $\mu$moles.g$^{-1}$, stress effect $F_{2,100} = 12.9$, $P = 0.00001$). However, as indicated by the interaction between diet and stress ($F_{2,100} = 3.2$, $P = 0.044$), the decline following escape response was not significant in the control diet, as baseline initial levels were lower than in experimental diet. Moreover, the triple interaction involved salinity ($F_{2,100} = 3.5$, $P = 0.036$), and indicated that for individual mean comparisons, the

**Table 6  Biochemical levels (means ± standard error) in hemolymph of *L. vannamei* reared at 37 and 6 psu (salinity(S)), fed with control and experimental (Exp) diets (D) for 6 weeks and exposed to hypoxia and escape response (st = stress test).**

|  |  | 37 psu |  | 6 psu |  | S | D | st | SxD | Sxst | Dxst | Trt |
|---|---|---|---|---|---|---|---|---|---|---|---|---|
|  |  | *Control* | *Exp* | *Control* | *Exp* |  |  |  |  |  |  |  |
| HEPATOPANCREAS | Total proteins (mg g$^{-1}$) |  |  |  |  | NS | NS | NS | * | ** | NS | NS |
|  | Baseline | 283 ± 18 | 267 ± 22 | 362 ± 43 | 258 ± 33 |  |  |  |  |  |  |  |
|  | Hypoxia | 337 ± 27 | 351 ± 42 | 258 ± 29 | 246 ± 17 |  |  |  |  |  |  |  |
|  | Escape | 225 ± 10 | 285 ± 19 | 369 ± 28 | 294 ± 30 |  |  |  |  |  |  |  |
|  | Total lipids (mg g$^{-1}$) |  |  |  |  | ** | NS | NS | ** | NS | NS | * |
|  | Baseline | 206 ± 22[abc] | 184 ± 24[abc] | 206 ± 50[abc] | 169 ± 27[abc] |  |  |  |  |  |  |  |
|  | Hypoxia | 257 ± 17[c] | 183 ± 13[abc] | 126 ± 18[ab] | 179 ± 13[abc] |  |  |  |  |  |  |  |
|  | Escape | 234 ± 10[bc] | 165 ± 14[abc] | 119 ± 34[a] | 191 ± 30[abc] |  |  |  |  |  |  |  |
|  | Triglycerides (mg g$^{-1}$) |  |  |  |  | ** | ** | NS | ** | NS | NS | NS |
|  | Baseline | 56 ± 5 | 53 ± 7 | 27 ± 6 | 49 ± 7 |  |  |  |  |  |  |  |
|  | Hypoxia | 53 ± 3 | 68 ± 3 | 29 ± 7 | 61 ± 3 |  |  |  |  |  |  |  |
|  | Escape | 55 ± 3 | 60 ± 4 | 30 ± 10 | 61 ± 7 |  |  |  |  |  |  |  |
| MUSCLE | Total proteins (mg g$^{-1}$) |  |  |  |  | NS | NS | ** | NS | NS | NS | NS |
|  | Baseline | 504 ± 57 | 560 ± 32 | 548 ± 37 | 584 ± 25 |  |  |  |  |  |  |  |
|  | Hypoxia | 643 ± 27 | 656 ± 22 | 593 ± 27 | 608 ± 10 |  |  |  |  |  |  |  |
|  | Escape | 693 ± 20 | 727 ± 16 | 694 ± 69 | 705 ± 47 |  |  |  |  |  |  |  |
|  | Lactate (mg g$^{-1}$) |  |  |  |  | ** | ** | ** | * | NS | NS | NS |
|  | Baseline | 5.9 ± 0.4 | 6.2 ± 0.8 | 3.8 ± 0.4 | 5.7 ± 0.4 |  |  |  |  |  |  |  |
|  | Hypoxia | 6.6 ± 0.6 | 7.8 ± 0.4 | 4.1 ± 0.7 | 5.3 ± 0.5 |  |  |  |  |  |  |  |
|  | Escape | 7.8 ± 0.4 | 8.5 ± 0.5 | 5.4 ± 0.6 | 8.2 ± 0.7 |  |  |  |  |  |  |  |
|  | Total lipids (mg g$^{-1}$) |  |  |  |  | NS | ** | ** | NS | NS | NS | NS |
|  | Baseline | 26.2 ± 5.2 | 42.0 ± 4.9 | 26.4 ± 2.7 | 35.9 ± 4.0 |  |  |  |  |  |  |  |
|  | Hypoxia | 37.9 ± 1.8 | 29.1 ± 4.9 | 36.4 ± 3.2 | 43.9 ± 4.4 |  |  |  |  |  |  |  |
|  | Escape | 43.5 ± 6.0 | 49.1 ± 2.3 | 45.3 ± 4.0 | 57.6 ± 6.3 |  |  |  |  |  |  |  |

**Notes.**

Results of three-way ANOVA are indicated in last columns (*, $P < 0.05$; **, $P < 0.01$; NS, not significant). Only when significant triple interaction was significant, values with different letters in the same row present significant differences ($P < 0.05$), following Tukey's HSD test.

influence of escape response was significant in the combination of 6 psu salinity and experimental diet only (Table 7). Similarly, AEC was higher with the experimental diet ($0.82 \pm 0.01$) than the control ($0.71 \pm 0.01$), given the higher content of ATP ($7.1 \pm 0.2$ vs. $5.1 \pm 0.2$ µmoles.g$^{-1}$), but also to lower content of ADP ($2.6 \pm 0.1$ vs. $3.3 \pm 0.1$ µmoles.g$^{-1}$) and AMP ($0.47 \pm 0.05$ vs. $1.15 \pm 0.08$ µmoles.g$^{-1}$) when compared to control diet (diet factor, $F_{1,102} = 61.9$, $P < 0.00001$, $F_{1,102} = 53.9$, $P < 0.00001$, $F_{1,102} = 23.5$, $P = 0.00004$, $F_{1,102} = 59.5$, $P = 0.00000$, respectively). The total adenylic nucleotide (TAN) concentration was also higher in shrimp fed experimental diet ($10.2 \pm 0.2$ µmoles.g$^{-1}$) when compared to control diet ($9.6 \pm 0.17$ µmoles.g$^{-1}$, diet effect $F_{1,102} = 6.7$, $P = 0.011$). In addition, ADP, AMP and TAN levels were lower at 37 psu ($2.7 \pm 0.1$, $0.68 \pm 0.06$ and $9.3 \pm 0.17$ µmoles.g$^{-1}$, respectively) than at 6 psu ($3.2 \pm 0.1$, $0.94 \pm 0.09$ and $10.6 \pm 0.2$ µmoles.g$^{-1}$, salinity effect $F_{1,102} = 8.3$, $P = 0.0048$, $F_{1,102} = 8.7$, $P = 0.004$ and $F_{1,102} = 6.7$, $P = 0.011$, respectively). However, the significant interaction found between diet and

**Table 7** Concentration of Arg-P, nucleotides and Adenylate energy charge (AEC) (means ± standard error) in muscle of *L. vannamei* reared at 37 and 6 psu (salinity (S)), fed with control and experimental (Exp) diets (D) for 6 weeks and exposed to hypoxia and escape.

| | 37 psu | | 6 psu | | S | D | st | SxD | Sxst | Dxst | Trt |
|---|---|---|---|---|---|---|---|---|---|---|---|
| | *Control* | *Exp* | *Control* | *Exp* | | | | | | | |
| Arg-P ($\mu$moles g$^{-1}$) | | | | | NS | ** | ** | NS | NS | * | * |
| Baseline | 2.51 ± 1.04[abc] | 5.93 ± 1.37[bcd] | 2.97 ± 0.68[abc] | 6.63 ± 1.20[cd] | | | | | | | |
| Hypoxia | 1.22 ± 0.43[a] | 8.24 ± 1.64[d] | 1.51 ± 0.40[a] | 3.05 ± 0.68[abc] | | | | | | | |
| Escape | 1.02 ± 0.24[a] | 2.14 ± 0.48[ab] | 0.62 ± 0.19[a] | 2.08 ± 0.32[ab] | | | | | | | |
| AMP ($\mu$moles g$^{-1}$) | | | | | ** | ** | NS | * | NS | NS | NS |
| Baseline | 1.0 ± 0.16 | 0.38 ± 0.04 | 1.34 ± 0.19 | 0.48 ± 0.10 | | | | | | | |
| Hypoxia | 1.11 ± 0.16 | 0.34 ± 0.06 | 1.32 ± 0.24 | 0.51 ± 0.14 | | | | | | | |
| Escape | 0.71 ± 0.10 | 0.60 ± 0.19 | 1.54 ± 0.27 | 0.55 ± 0.12 | | | | | | | |
| ADP ($\mu$moles g$^{-1}$) | | | | | ** | ** | NS | NS | NS | NS | NS |
| Baseline | 2.97 ± 0.30 | 2.26 ± 0.13 | 3.51 ± 0.26 | 2.48 ± 0.21 | | | | | | | |
| Hypoxia | 3.47 ± 0.22 | 2.07 ± 0.19 | 3.44 ± 0.24 | 3.01 ± 0.29 | | | | | | | |
| Escape | 3.22 ± 0.18 | 2.72 ± 0.26 | 3.42 ± 0.43 | 3.33 ± 0.21 | | | | | | | |
| ATP ($\mu$moles g$^{-1}$) | | | | | NS | ** | NS | NS | NS | NS | NS |
| Baseline | 4.61 ± 0.32 | 7.07 ± 0.30 | 5.03 ± 0.26 | 7.54 ± 0.70 | | | | | | | |
| Hypoxia | 5.19 ± 0.41 | 6.58 ± 0.52 | 5.69 ± 0.61 | 7.63 ± 0.60 | | | | | | | |
| Escape | 5.52 ± 0.37 | 6.21 ± 0.44 | 4.67 ± 0.60 | 7.78 ± 0.40 | | | | | | | |
| TAN ($\mu$moles g$^{-1}$) | | | | | ** | * | NS | * | NS | NS | NS |
| Baseline | 8.6 ± 0.5 | 9.7 ± | 9.9 ± 0.4 | 10.5 ± 0.7 | | | | | | | |
| Hypoxia | 9.8 ± 0.4 | 9.0 ± 0.6 | 10.4 ± 0.4 | 11.1 ± 0.5 | | | | | | | |
| Escape | 9.5 ± 0.3 | 9.5 ± 0.2 | 9.6 ± 0.4 | 11.7 ± 0.3 | | | | | | | |
| AEC (ATP+1/2ADP/ATP+ADP+AMP) | | | | | NS | ** | NS | NS | NS | NS | NS |
| Baseline | 0.71 ± 0.03 | 0.84 ± 0.01 | 0.69 ± 0.02 | 0.83 ± 0.02 | | | | | | | |
| Hypoxia | 0.71 ± 0.03 | 0.85 ± 0.02 | 0.70 ± 0.03 | 0.82 ± 0.03 | | | | | | | |
| Escape | 0.75 ± 0.02 | 0.79 ± 0.03 | 0.66 ± 0.04 | 0.81 ± 0.02 | | | | | | | |

**Notes.**
Results of three-way ANOVA are indicated in last columns (*, $P < 0.05$; **, $P < 0.01$; NS, not significant). Only when significant triple interaction was significant, values with different letters in the same row present significant differences ($P < 0.05$), following Tukey's HSD test.

TAN, total adenylic nucleotides; AEC, adenylic energy charge.

salinity indicated that salinity effects was observed only for the control diet in the case of AMP and for the experimental diet in the case of TAN (Table 7, interaction salinity x diet, $F_{1,102} = 4.5$, $P = 0.036$ and $F_{1,102} = 4.1$, $P = 0.047$, respectively). Finally, AEC and levels of nucleotides did not show any significant differences during acute stress (hypoxia or escape).

# DISCUSSION

According to this research, is possible to increase shrimps energy status (AEC and Arg-P) by means of dietary manipulation which also resulted in potential improvement of growth performance and modulation of stress response of *L. vannamei* to multiple stressors (low salinity, hypoxia and manipulation).
## Productive performance at low salinity and diet

Despite that *L. vannamei* is considered a euryhaline shrimp with a tolerance to salinity ranging 1 to 50 psu (*Pante, 1990*), previous literature and our study indicate that growth performance and survival is affected at low salinity due to the high energy requirements, mainly for osmoregulation (for reviews see *Romano & Zeng, 2012*; *Li et al., 2017*). However, the majority of studies were carried out at salinities below 5 psu, while this work considered 6 psu in order to simulate the typical salinity of well water in Baja California Sur. The latter is still a suboptimal salinity given that growth and survival were lower under this condition when shrimp were fed control diet. In contrast, the experimental diet (in spite of low protein level) improved overall performance in culture, as dietary manipulation of macro and micronutrients enhances growth performance and physiological adaptation of *L. vannamei* at low salinity, as previously reported (*Gong et al., 2004*; *Roy, Davis & Saoud, 2006*; *Wang et al., 2014*; *Xu et al., 2017a*), and discussed below.

Protein is the most important dietary component; therefore, it has received special attention as strategy to enhance growth and survival under the assumption that this energy source improves osmoregulation effectively (*Romano & Zeng, 2012*). However, dietary protein requirements at low salinity and their role as enhancers of growth and survival remains controversial (*Li et al., 2017*). Survival at low salinity decreases with increasing levels of proteins in the diet (*Wang, Ma & Dong, 2005*; *Zhu et al., 2010*; *Wang et al., 2015*). This statement is in accordance with our study, since the lowest performance in terms of growth and survival was obtained at 6 psu with the control diet (higher protein content). In contrast, several researches obtained growth improvement at low salinity with higher protein content in the diet (*Liu et al., 2005*; *Li et al., 2007*; *Li et al., 2011*), while other reports indicate that this is not always the case (*Zhu et al., 2010*).

In the present work, other nutrients besides protein might have contributed to the high performance observed with the experimental diet. The difference of protein content (14%) between both diets is compensated by a slightly higher lipid (1%; Table 3) content and a considerably higher carbohydrate (CBH) content (more than 10% if estimated on the basis of nitrogen free extract; Table 3) in the experimental diet. In addition to performance, biochemical indicators of the beneficial role of the experimental diet were also observed, since despite the low content of gross energy, levels of energy-phosphorylated compounds such as Arg-P and ATP in shrimp muscle were higher. Thus, this potentialized availability of cell-energy can be attributed to an ostensibly better allocation of energy from lipids and CBH for osmoregulation, as discussed in previous reviews on shrimp culture at low salinities (*Romano & Zeng, 2012*; *Chen et al., 2014*; *Wang et al., 2014*; *Li et al., 2017*). This explains the higher protein-efficiency ratio found with the experimental diet, especially at low salinity, indicating a sparing of protein for growth, while CBH and lipids are used for metabolism. In accordance, proteomic analyses have revealed that glycometabolism (tricarboxylic acid cycle, glycolysis, and gluconeogenesis) in the hepatopancreas is enhanced at 3 psu (*Xu et al., 2017b*). Moreover, a differential use of energy substrates for osmoregulation in relation to the relative levels of protein and CBH in diet were detected for the crab, *Chasmagnathus granulate*, a good-eurhyaline osmoregulator (*Da Silva & Kucharski, 1992*).

In a study of CBH in diets, improved performance (growth and survival) at low salinity was detected with intermediate levels of digestible CBH (20% starch) in iso-proteinic (40%) and iso-lipidic (6%) diets, ranging from 5 to 30% CBH (*Wang et al., 2014*). According to estimations of energy budget, when CBH levels increase in a diet, there is a concomitant reduction of protein levels, with optimal CBH level between 26 and 30% for maximum growth at low salinities (1 to 8 psu) that corresponds to an increased energy destined for growth (*Wang et al., 2014*). In a recent study, *Wang et al. (2015)* used iso-energetic diets with different protein:CBH ratios of 26–38%: 30–14% and the highest growth at 3 psu was observed with a diet consisting of 19% CBH and 34% protein. Inclusion of CBH in diets deserves particular consideration since shrimp are supposedly not metabolically adapted to high CBH levels. Indeed, a limit of 33% was suggested according to the starch-digestion capacity by $\alpha$-amylase and saturation of glycogen in the digestive gland (*Rosas et al., 2002*). Moreover, the metabolic saturation of the capacity to use CBH is more notorious in farmed and genetically selected shrimp when compared to wild populations. A reduction of allele frequency in amylase genes of domesticated shrimp (25th generation) was related to a reduced ability of shrimp to use dietary CBH (*Arena et al., 2003*). Similarly, the 7 [th] generation of cultured shrimp (*L. vannamei*) presented high dependence of protein for metabolism and immune response (*Pascual et al., 2004*). This explains why intermediate (19–30%) CBH levels in diets are optimal at low salinity (*Wang, Ma & Dong, 2005*; *Wang et al., 2014*; *Wang et al., 2015*). However, these results depend on the particular nature of each selection program with regard to the development of culture over generations and specific culture conditions. Contrary to the conditions employed by *Arena et al. (2003)* and *Pascual et al. (2004)*, in our study, shrimp were obtained from Acuacultura Mahr, this enterprise has a genetic selection program in which biofloc hyper-intensive culture conditions and low protein levels in the diet are gradually implemented over successive generations. Therefore, the threshold for maximum CBH can be increased through selection, and these shrimps were best adapted to high CBH in the diet. The low growth with high level of dietary CBH is partially attributed to carbohydrolases-shrimp deficiency, thus, it was suggested that probiotic bacteria should be included in feed to improve CBH digestibility (*Olmos et al., 2011*). In our study, the experimental diet included a probiotic mix that could have contributed to an enhanced CBH assimilation, given that it contained yeast, bacilli and lactobacilli with CBH-processing capabilities (data not part of this research).

In addition, despite the difference of lipid levels between diets was low (1%), triglyceride levels in the hepatopancreas were higher in shrimp cultured at low salinity and fed experimental diet (Table 6). The influence of total and specific lipid levels in diet was analyzed in relation to performance at low salinity. In regard to total lipids, intermediate levels of 9% (range: 6–12%) resulted in the highest growth at 2 psu, which can be partially attributed to a protein sparing effect, as indicated by the low glutamic oxaloacetic transaminase (*Xu et al., 2017a*). According to lipid levels and enzyme activities, *Chen et al. (2014)* determined that lipid metabolism provides enough energy to osmoregulate efficiently at low and high salinities, improving performance during culture. The level of several specific lipids such as cholesterol, phospholipids and highly unsaturated fatty acids

(HUFA) was also analyzed in relation to performance and osmoregulatory capacities at low salinity (for reviews see *Palacios & Racotta, 2007*; *Romano & Zeng, 2012*; *Li et al., 2017*). Apparently, in our study, the beneficial effect of the experimental diet at low salinity is not related with higher HUFA levels, given that the opposite difference was detected: lower proportion of individual (e.g., $20{:}6n-3$) and total HUFA.

The beneficial effect of the experimental diet analyzed in this research was not only related to the low total protein content and high CBH/lipid level, but also to individual amino acids and other nutrients that were included. A diet supplemented with essential amino acids in their free form is a successful strategy to reduce pressure of protein demand as energy source (*Claybrook, 1983*). For example, isoleucine, leucine, threonine, and *L*-tryptophan added in their crystalline form, improve performance at low salinity (*Li et al., 2017*). Inclusion of free amino acids can replace fish meal; hence, these were included in the experimental diet, decreasing protein costs and environmental deterioration, as suggested previously (*Huai et al., 2009*; *Xie et al., 2015*). With regard to other nutrients, the osmoregulatory ability of shrimp improves by incorporating astaxanthin (*Flores et al., 2007*), vitamins and minerals (*Gong et al., 2004*), phosphorus (*Cheng et al., 2006*), and microorganisms (*Avnimelech, 2012*) in diets. Therefore, these micronutrients were added in the experimental diet, which explains our results, although it is not possible to identify the exact nutrient or set of nutrients responsible for the improved performance at low salinity. We decided to elaborate this diet in order to evaluate the synergic effect of all the components (low protein, free amino acids, vitamins, minerals and probiotic mixture), given that individual effects are well documented, but studies considering two or more components are scarce.

## Stress response

The main findings of this study must be evaluated in relation to two key concepts of stress physiology. First, energy-limiting stress tolerance, mostly applied to chronic stress during overall fitness (growth and reproduction), which is limited or suppressed due to the re-allocation of energy to cope with stress (*Sokolova, 2013*). As discussed earlier, this was the case for the limited growth at low salinity, although no evidence of limited energy was observed in terms of the variables measured, reserves, and cellular energy charge, including Arg-P levels. Second, increased vulnerability to a second stressor (in this study hypoxia or escape), applied in shrimp that were already exposed to a first stressful situation (low salinity in our research) (*Chrousos & Gold, 1992*). Hence, it was important to perform a comprehensive analysis that addressed the particular effects of acute stressors and the level of response in relation to salinity conditions, as well as the influence of diet. Under this context, the exposure to stressors entails an increase in energy demand at the cellular level, thus, shrimp fitness depends on their capacity to allocate energy, which is constrained by the physiological and nutritional state (*Tseng & Hwang, 2008*).

The main response to hypoxia exposure (regardless of salinity and diet) was an increase in hemolymph glucose and lactate, in accordance with the activation of anaerobic metabolism during oxygen shortage (*Gäde, 1984*; *Abe, Hirai & Okada, 2007*; *Soñanez Organis, Racotta*
& *Yepiz-Plascencia, 2010*). An intense muscular activity is also related to the anaerobic-metabolism activation caused by an insufficient delivery of oxygen to tissues, with its concomitant glucose and lactate increase (*Yu et al., 2009*; *Robles-Romo, Zenteno-Savín & Racotta, 2016*). However, in this work, glucose did not increase right after tail flipping, as detected by *Robles-Romo, Zenteno-Savín & Racotta (2016)*, where such response was not appreciated immediately, but only after one hour. In addition, lactate response was considerably higher with hypoxia when compared to tail flipping, this can be attributed to the duration of the stress (less than one minute for tail flipping and 30 min for hypoxia), also taking into account that the accumulation of lactate in hemolymph was clearly associated with the duration of both stressors (*Gäde, 1984*). In contrast, a decrease in protein levels was observed following escape, this effect had not been observed in previous studies (*Robles-Romo, Zenteno-Savín & Racotta, 2016*); however, it could be related to the usage of circulating protein to satisfy muscle energy demand, as observed during swimming (*Duan et al., 2014*).

Decreased AEC and Arg-P are also typical metabolic responses associated with the increase in energy demands during tail flipping (*Onnen & Zebe, 1983*; *England & Baldwin, 1983*; *Gäde, 1984*; *Thebault et al., 1994*; *Robles-Romo, Zenteno-Savín & Racotta, 2016*) and oxygen shortage due to ATP synthesis during hypoxia (*Gäde, 1984*; *Abe, Hirai & Okada, 2007*; *Sokolova, 2013*). AEC did not present significant differences in the experimental diet at 6 psu during acute stress (hypoxia or escape), in fact, only mild AEC decrease was observed after 6 h under hypoxia (*Abe, Hirai & Okada, 2007*) and 20 s of tail flipping (*Robles-Romo, Zenteno-Savín & Racotta, 2016*).

Several interesting findings were observed regarding response to acute stressors in relation to diet and salinity. The increase in hemolymph glucose during hypoxia was more notorious at 37 than at 6 psu, most probably due to the low capacity of response to a second stressor in shrimp that had been submitted to a first chronic stressor (low salinity) for several weeks. Similarly, protein in hepatopancreas increased during hypoxia at 37 psu only. As hemocyanin is synthesized in the hepatopancreas (*Senkbeil & Wriston Jr, 1981*), an increase in protein levels might correspond to the synthesis of this or other proteins involved in hypoxic response (e.g., heat shock proteins) (*De La Vega et al., 2006*) or hypoxia-inducible factor (HIF) (*Soñanez Organis, Racotta & Yepiz-Plascencia, 2010*). Such a putative adaptative response was mitigated in shrimp under low salinity, i.e., a previous stress condition increased vulnerability to a subsequent stressor. Another result that can be related to the dual stress exposure is the significantly lower lipid level found in hepatopancreas of shrimp fed control diet at 6 psu under both acute stressors (hypoxia and escape response). This triple interaction indicates that with stress as a single factor, lipids are not mobilized to satisfy the energy demand associated with muscular activity, contrary to dual stress. In turn, lipid mobilization does not occur with higher CBH content in the diet.

In regard to the interaction between stress and diet, the increase in hemolymph glucose found in shrimp exposed to hypoxic stress was more pronounced when fed experimental diet, probably due to the high CBH level in the experimental diet, as discussed earlier. Previous studies observed higher glucose levels in shrimp fed high CBH content; however,

this was detected in wild shrimp and no stressor was applied (*Arena et al., 2003*). As mentioned earlier, Arg-P was higher in shrimp fed experimental diet and decreased to provide energy for tail flipping. In shrimp fed with the experimental diet, this indicates a higher energy availability to cope with acute stress. Therefore, the resulting levels of Arg-P after tail flipping were lower for shrimp fed control diet; however, the average Arg-P consumption was higher for the experimental diet ($4.2 \ \mu\text{mol g}^{-1}$) when compared to the control ($1.95 \ \mu\text{mol g}^{-1}$). Moreover, the decrease was more notorious at 6 psu in shrimp fed control diet, which suggests that the experimental diet also copes with dual stress condition.

## CONCLUSIONS

According to this research, diet can increase energy status (AEC and Arg-P) to successfully overcome potential multifactorial stressors, which are common in farming systems. Exposure to chronic low salinity showed no evidence of limited energy in terms of energy reserves at the cellular level, in contrast to acute stress by hypoxia or escape response, which imposed high energy demands that depended on diet and the previous condition of chronic stress of low salinity.

## ACKNOWLEDGEMENTS

The authors are grateful to Sandra de la Paz for her support the experimental facilities; Ernesto Goytortúa Bores for diet preparation; and Paola Magallón Servín, Melisa López Vela, Edgar Salvador Reyes and María Cristina Galaviz who designed, formulated and provided the probiotic mixture; María Dolores Rondero Astorga for proximate analyses of diets; Roberto Hernández-Herrera and Rosalinda Salgado for technical assistance in biochemical analyses and adenylic nucleotides, respectively; and María Olivia Arjona López for fatty acids analyses.

### Funding

This research was funded by CONACYT (grant number 260464), CONACYT (FINNOVA) and DACI-Conacyt 296397. The funders had no role in study design, data collection and analysis, decision to publish, or preparation of the manuscript.

### Grant Disclosures

The following grant information was disclosed by the authors:
CONACYT: grant number 260464.
CONACYT (FINNOVA).
DACI-Conacyt: 296397.

### Competing Interests

The authors declare there are no competing interests.

## Author Contributions

- Eliza M. Martínez-Antonio conceived and designed the experiments, performed the experiments, analyzed the data, prepared figures and/or tables, authored or reviewed drafts of the paper, approved the final draft.
- Ilie S. Racotta conceived and designed the experiments, analyzed the data, contributed reagents/materials/analysis tools, prepared figures and/or tables, authored or reviewed drafts of the paper, approved the final draft.
- Juan C. Ruvalcaba-Márquez conceived and designed the experiments, performed the experiments, approved the final draft.
- Francisco Magallón-Barajas conceived and designed the experiments, analyzed the data, contributed reagents/materials/analysis tools, prepared figures and/or tables, approved the final draft.

## Data Availability

Raw data is provided in the Supplemental Files.

## Supplemental Information

Supplemental information for this article can be found online at http://dx.doi.org/10.7717/peerj.6850#supplemental-information.

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
