# Peer review of "Modulation of stress response and productive performance of *Litopenaeus vannamei* through diet"

_PeerJ, doi:10.7717/peerj.6850_

## Round 0.1 · original submission · Major Revisions

Please provide a point-by-point response to each of the reviewers' comments, noting in particular reviewer 1's concerns about lack of detail provided and reviewer 2's concerns regarding the statistical analyses.

Reviewer 1 ·

Basic reporting

This Manuscript needs to be improved before publication: some suggestions are listed below

There is no information in introduction about type of diet (composition) which has already evaluated on shrimp cultured in or submitted to low salinity condition. This would be useful to introduce the choice of experimental diet composition used in this study.
There is also no mention of probiotic interest in diet to face stressful conditions.
The use of “global means” all along the “Result” part does not facilitate the comprehension. The readers have to calculate all cited values (as these mean values are not include in the corresponding tables) to understand what the authors mean.

Experimental design

Some important details on the experimental conditions are missing in the text (the supplementary data were useful to find these missing details in manuscript):

#Line 82-92: Initial shrimp weight at the beginning of differentiate feeding period; The number of replicate per treatment (salinity and diet); the number of shrimp used (total and per treatment)

#Line 101-106 and Table 1: More details on diets composition and ingredients are required : origin and proportion of carbohydrates, origin and concentration of probiotic and yeast (commercial or lab strain, genre and species if known), different mix composition). The diet formulae are not complete, sum of ingredients cited below 100%. What was the percentage of free amino acids in diets?

#Line 128-131: How long after stress, are the shrimp harvested and sampled?

Validity of the findings

The authors showed that an adapted diet improve survival, protein conversion rate and productive performance when cultured at 6psu. The shrimp capacity to face stressful condition is also improved by this adapted nutrition: the shrimp energy status is improved after 42 days of culture whatever the salinity

Parts of "Discussion" have to be reviewed :
#Line 325-352: This part of the discussion on the CBH and its impact on productivity is a bit long, while there is no information on CBH level in the experimental diet of this study. This part should be reduced if additional data on the diet formula is not added in Material and Methods (cf previous comment).
#Line 352-354 : Even if « Probiotic » is a small part of the experimental diet, it would be interesting to specify in the Mat & Max, what type of bacteria and yeasts were added and for which purpose. Especially if these microorganisms could have contributed to assimilate CBH.

Data about survival to additionals acutes stresses (escape and hypoxia) could complete interestingly the results obtained in this study.

Additional comments

Pay attention: "psu" in not written correctly in different places; “37 psu” is sometimes replaced by “35 psu”
Line 167: “growth of” can be deleted

Reviewer 2 ·

Basic reporting

The present manuscript is well structured, the objective is clear, English is well written, the bibliography is adequate, I suggest that some of the tables could be transformed into a figure to facilitate the visualization of the results.

Experimental design

The present research is original because it integrates the effect of environmental stressors with diet and in response it measures several biochemical and aquacultural parameters. The research question is well defined. The methodology used is described in sufficient detail and can be replicated. Also in the part and results it is suggested to group them as they are too long and it is tedious to follow them.

Validity of the findings

The literature cited is considerad that adequate and covers the subject treated. The impact of this research is related to shrimp management from the aquuacultural point of view by the focus of the evaluated parameters. In the part of the statistical analysis I suggest the authors use the arcsine transformation in the parameters that they describe and that are in percentages because to apply the ANOVA in that type of data they must be transformed by that function since otherwise they would be violated the assumptions with which the statistical test works. The conclusions are considered adequate in relation to the results obtained

Additional comments

In the part of the statistical analysis I suggest the authors use the arcsine transformation in the parameters that they describe and that are in percentages because to apply the ANOVA in that type of data they must be transformed by that function since otherwise they would be violated the assumptions with which the statistical test works.
Also in the part of results it is suggested to group them because as they are presented they are too long and sometimes repetitive.
On line 330 it is suggested to change the prayer to. In a recent study Wang et al (2015) used isoenergetics diets with different protein: CBH ratios of 26-30%: 30-14% were compared with the highest growth at 3 psu was observed with a diet consisting of 19% CBH and 34% protein.

---

## Round 0.2 · accepted · Accept

Thank you for addressing the reviewers' comments, which I believe have improved the manuscript considerably.

#